# IGF-1-enhanced miR-513a-5p signaling desensitizes glioma cells to temozolomide by targeting the NEDD4L-inhibited Wnt/β-catenin pathway

**Ku-Chung Chen** [1,2☯], **Peng-Hsu Chen**[1,2☯], **Kuo-Hao Ho**[1,2☯], **Chwen-Ming Shih**[1,2], **Chih-Ming Chou**[1,2], **Chia-Hsiung Cheng**[1,2], **Chin-Cheng Lee**[3]*

**1** Graduate Institute of Medical Sciences, College of Medicine, Taipei Medical University, Taipei, Taiwan, **2** Department of Biochemistry and Molecular Cell Biology, School of Medicine, College of Medicine, Taipei Medical University, Taipei, Taiwan, **3** Department of Pathology and Laboratory Medicine, Shin Kong Wu Ho-Su Memorial Hospital, Taipei, Taiwan

☯ These authors contributed equally to this work.
* cclee6666@yahoo.com.tw

**Data Availability Statement:** All relevant data are within the manuscript and its Supporting Information files.

## Abstract

Temozolomide (TMZ) is a first-line alkylating agent for glioblastoma multiforme (GBM). Clarifying the mechanisms inducing TMZ insensitivity may be helpful in improving its therapeutic effectiveness against GBM. Insulin-like growth factor (IGF)-1 signaling and micro (mi)RNAs are relevant in mediating GBM progression. However, their roles in desensitizing GBM cells to TMZ are still unclear. We aimed to identify IGF-1-mediated miRNA regulatory networks that elicit TMZ insensitivity for GBM. IGF-1 treatment attenuated TMZ cytotoxicity via WNT/β-catenin signaling, but did not influence glioma cell growth. By miRNA array analyses, 93 upregulated and 148 downregulated miRNAs were identified in IGF-1-treated glioma cells. miR-513a-5p from the *miR-513a-2* gene locus was upregulated by IGF-1-mediated phosphoinositide 3-kinase (PI3K) signaling. Its elevated levels were also observed in gliomas versus normal cells, in array data of The Cancer Genome Atlas (TCGA), and the GSE61710, GSE37366, and GSE41032 datasets. In addition, lower levels of neural precursor cell-expressed developmentally downregulated 4-like (NEDD4L), an E3 ubiquitin protein ligase that inhibits WNT signaling, were found in gliomas by analyzing cells, arrays, and RNA sequencing data of TCGA glioma patients. Furthermore, a negative correlation was identified between miR-513a-5p and NEDD4L in glioma. NEDD4L was also validated as a direct target gene of miR-513a-5p, and it was reduced by IGF-1 treatment. Overexpression of NEDD4L inhibited glioma cell viability and reversed IGF-1-repressed TMZ cytotoxicity. In contrast, miR-513a-5p significantly affected NEDD4L-inhibited WNT signaling and reduced TMZ cytotoxicity. These findings demonstrate a distinct role of IGF-1 signaling through miR-513a-5p-inhibited NEDD4L networks in influencing GBM's drug sensitivity to TMZ.

**Funding:** This study was sponsored by the Shin Kong Wu Ho-Su Memorial Hospital, Taiwan (SKH-8302-106-DR-30) to C-CL, and grants from the Ministry of Science and Technology, Taiwan (MOST 106-2320-B-038-051-MY3) to K-CC, Ministry of Education, Taiwan (DP2-107-21121-01-NK and DP2-108-21121-01-NK) to C-MS, and Taipei Medical University-Shin Kong Wu Ho-Su Memorial Hospital, Taiwan (intramural grant no. SKH-TMU-107-03) to K-CC. The funders had no role in study design, data collection and analysis, decision to publish, or preparation of the manuscript.

**Competing interests:** The authors have declared that no competing interests exist.

## Introduction

Glioblastoma multiforme (GBM) belongs to grade IV primary malignant gliomas with poor prognoses and high lethality in adults [1, 2]. Several tumor microenvironmental factors were identified to enhance the risk of brain tumors, including the insulin-like growth factor (IGF) signaling axis [3]. When the circulating IGF-1 ligand binds to its receptor, IGF-1R, this tyrosine kinase receptor is activated through an autophosphorylation mechanism. Subsequently, two major downstream pathways, phosphoinositide 3-kinase (PI3K)/AKT and Ras/extracellular signal-regulated kinase (ERK) pathways, are enhanced to prevent cell death or promote cell growth. In gliomas, IGF-1 modulates cell proliferation and strongly stimulates cell migration [4]. IGF-1 also regulates inflammatory responses in glioma cells via influencing hypoxia-inducible factor (HIF)-1α-toll-like receptor 9 (TLR9) cross talk [5]. Furthermore, increasing evidence suggests that IGF-1 signaling is involved in drug resistance mechanisms, resulting in glioma progression [6]. The IGF-1/IGF-1R axis was identified to underlie resistance to colony-stimulating factor-1 receptor (CSF-1R) inhibition in gliomas [7]. By increasing Bcl-2 expression and decreasing caspase-3 protease activity, IGF-1 significantly decreased the etoposide-induced apoptosis of glioma cells [8]. Taken together, comprehensively investigating IGF-1-mediated gene networks may be helpful in understanding the progression of gliomagenesis and provide innovative therapeutic strategies for glioblastomas.

Micro (mi)RNAs are endogenous, small, non-coding RNAs that inhibit gene expressions by binding to the 3' untranslated region (UTR) of their target messenger (m)RNAs. Aberrant miRNA expressions were identified in GBM development [9]. For example, miR-10b, a highly expressed onco-miR in all GBM subtypes, was suggested as being a potential target for GBM therapy [10]. Elevation of miR-215 levels by hypoxia is necessary for reprogramming glioma-initiating cells in GBM occurrence and recurrence [11]. miR-513a-5p, an intergenic miRNA, comes from two different gene loci: miR-513a-1 and miR-513a-2. The roles of miR-513a in tumorigenesis are still unclear, especially in GBM. Only one study reported that upregulated miR-513a-5p levels were observed in GBM patients compared to controls [12]. The functions and molecular mechanisms of miR-513a-5p in glioma progression need to be further studied.

Neural precursor cell-expressed developmentally downregulated 4-like (NEDD4L, also known as NEDD4-2) is an E3 ubiquitin protein ligase belonging to the NEDD4 family and contains a homologous E6-associated protein C-terminus (HECT) domain [13]. The best known function of NEDD4L is as an ion channel regulator, including the epithelial sodium channel (ENaC) [14], Na$^+$-Cl$^-$ cotransporter (NCC) [15], voltage-gated sodium channels (Navs) [16], and so on. Recently, a role of NEDD4L in carcinogenesis was identified. NEDD4L negatively regulates canonical WNT signaling in colorectal cancer [17]. Decreased NEDD4L levels were correlated with poor prognoses in gastric cancer patients [18]. Similarly, in gliomas, reduced NEDD4L expression was associated with aggressive progression and worse prognoses [19], suggesting that NEDD4L could be a tumor suppressor that inhibits tumorigenesis. However, no other studies have reported the molecular mechanisms regulating NEDD4L expression during cancer progression.

Temozolomide (TMZ), an alkylating agent of the imidazotetrazine series, is a first-line chemotherapeutic drug for clinically treating malignant gliomas. Nevertheless, gradually increasing drug resistance to TMZ has obviously decreased its therapeutic effects in glioma patients. A previous study suggested that blockade of the Hedgehog/GLI1-regulated IGF-1 pathways countered the intrinsic and acquired resistance of glioma stem cells to TMZ [20]. The mechanisms of IGF-1-influenced TMZ resistance in glioma cells are still unclear. Furthermore, no studies have reported if IGF-1-regulated miRNAs participate in drug resistance or glioma progression. In the present study, we attempted to identify IGF-1-influenced miRNA networks in

reducing glioma cell sensitivity to TMZ treatment. Using microarray and bioinformatic analyses, we found that miR-513a-5p was the most upregulated miRNA in IGF-1-treated U87-MG cells through PI3K pathways. NEDD4L, identified as a direct target gene of miR-513a-5p, was significantly downregulated in GBM patients and IGF-1-treated glioma cells. Overexpression of NEDD4L significantly reduced glioma cell viability. Finally, our results suggest that IGF-1-upregulated miR-513a-5p signaling attenuated glioma cell sensitivity to TMZ via inhibiting NEDD4L-inactivated WNT/β-catenin pathways.

## Materials and methods

### Chemicals and reagents

Human glioblastoma Hs-683, M059K, and U87-MG cells were purchased from the Bioresource Collection and Research Center (Hsinchu City, Taiwan). Primary human astrocytes were purchased from Thermo Fisher Scientific (Waltham, MA, USA). Other cell culture-related reagents were purchased from GIBCO-BRL (Grand Island, NY, USA). Anti-phosphorylated (p)-Ser9-GSK3β, GSK3β, p-β-catenin, and β-catenin antibodies were purchased from Cell Signaling Technology (Danvers, MA, USA). Anti-cyclin D1, NEDD4L, and β-actin antibodies were purchased from GeneTex (Hsinchu City, Taiwan). Temozolomide (cat. no. T2577), LY294002 (cat. no. L9908), U0126 (cat. no. U120), and 3-[4,5-dimethylthiazol-2-yl]-2,5-diphenyl tetrazolium bromide (MTT) (cat. no. M2128) were purchased from Sigma-Aldrich (St. Louis, MO, USA). Polyvinylidene difluoride (PVDF) membranes, an enhanced chemiluminescence (ECL) solution (cat. no. WBKLS0500), and the anti-p-Tyr216-GSK3b antibody were purchased from Millipore (Billerica, MA, USA). The human IGF-1 recombinant protein (cat. no. PHG0071), Trizol® reagent (cat. no. 15596026), Lipofectamine 3000 (cat. no. L3000015), and secondary antibodies were purchased from Invitrogen (Thermo Fisher Scientific). SYBR® Green PCR Master Mix (cat. no. 4309155), the MultiScribe (tm) Reverse Transcriptase Kit (cat. no. N8080234), TaqMan Advanced miRNA cDNA Synthesis Kit (cat. no. A28007), TaqMan® pri-miR-513a-1 (cat. no. Hs03296262_pri), pri-miR-513a-2 (cat. no. Hs03295531_pri), TaqMan® Advanced miR-513a-5p (cat. no. 479483_mir), and TaqMan® Advanced miR-191-5p (cat. no. 477952_mir) were purchased from Applied Biosystems (Thermo Fisher Scientific). The dual-luciferase reporter assay system (cat. no. E1910) was purchased from Promega (Madison, WI, USA). Glycogen synthase kinase 3β (GSK3β) inhibitor VIII was purchased from Cayman (Ann Arbor, MI, USA). Primer sets were synthesized by Genomics BioSci & Tech (Xizhi, New Taipei City, Taiwan). The short hairpin (sh) RNA was purchased from the National RNAi Core Facility (Nankang, Taiwan). TOPFlash reporter plasmids (M50 Super 8x TOPFlash) was purchased from addgene (Watertown, MA, USA). OxiSelect™ Oxidative DNA Damage ELISA Kits was purchased from CELL BIOLABS, INC. (San Diego, CA, USA). Unless otherwise specified, all other reagents were of analytical grade.

### Cell culture, treatments, and transfection

Minimum essential Eagle medium (MEM) was used to maintain U87-MG cells; a 1:1 mixture of Dulbecco's modified Eagle's medium (DMEM) and Ham's F12 medium with 2.5 mM L-glutamine was used to maintain M059K cells; DMEM with 4 mM L-glutamine was used to maintain HS-683 cells; and astrocyte cells were maintained in DMEM with N-2 Supplement. All cells were supplemented with 10% fetal bovine serum (Biological Industries, Cromwell, CT, USA), 100 units/ml penicillin, 100 μg/ml streptomycin, 1 mM sodium pyruvate, and 1 mM nonessential amino acids at 37°C in a 5% $CO_2$ incubator. For drug treatment, different doses of IGF-1 and TMZ were added to overnight-cultured cells for the indicated times. For

inhibitor treatment, 10 μM U0126 and 5 μM LY294002 were respectively pretreated for 1 h, and then 200 ng/ml IGF-1 was added to overnight-cultured cells for another 48 h. To conduct the transfection experiments, cells were seeded into a 12-well plate at a density of $10^5$ cells/well. After achieving 70% confluence in a well, indicated doses of pCDH-miR-513a-5p, pCDH-NEDD4L, 1 μg sh-β-catenin plasmids, and 500 ng pmiRGLO 3'UTR reporter plasmids were respectively transfected with Lipofectamine 3000 (Invitrogen) according to the manufacturer's instructions. After 24 h of incubation, cells were lysed for further study.

## Cell viability assay

MTT assay and cell counting assay with Trypan blue stain were respectively used to measure cell viability. Cells were seeded on a 96-well plate at $8 \times 10^3$ cells/well overnight, followed by treatment with various concentrations of IGF-1 or TMZ for another 48 h. Before the end of treatment, 0.5 mg/ml MTT was added to each well for 4 h. Supernatants were carefully aspirated, and formazan crystals were dissolved using dimethyl sulfoxide (DMSO). The absorbance was measured at 550 nm with a Thermo Varioskan Flash reader (Thermo Fisher Scientific). For cell counting assay, after cells were stained with Trypan blue, cell numbers were measured by manual cell counting using a hemocytometer.

## Immunoblot analysis

RIPA buffer (1% Nonidet P-40, 0.5% deoxycholate, and 0.1% sodium dodecylsulfate (SDS) in phosphate-buffered saline (PBS)) containing a protease inhibitor cocktail (Calbiochem, Billerica, MA, USA) was used to lyse cells, and then the mixture was centrifuged at 12,000 rpm for 10 min at 4˚C. The supernatant was used as the total cell lysate. Lysates were denatured in 2% SDS, 10 mM dithiothreitol, 60 mM Tris-hydrochloric acid (Tris-HCl, pH 6.8), and 0.1% bromophenol blue, and loaded onto a 10% or 15% polyacrylamide/SDS gels according to the target proteins with a molecular weight of >50 or <50 kDa. Separated proteins were then transferred onto a PVDF membrane. The membrane was blocked for 1 h at room temperature in PBS containing 5% nonfat dry milk and incubated overnight at 4˚C in PBS-T (PBS and Tween 20) containing the primary antibody. After washing, the membrane was incubated with the secondary antibody conjugated to horseradish peroxidase for 1 h at room temperature, and then washed with PBS-T again. The primary and secondary antibodies were respectively diluted to 1:1000 and 1:3000 with PBS-T buffer. An enhance chemiluminescence non-radioactive detection system was used to detect antibody-protein complexes. All the raw data were listed as S2 Fig.

## Microarray, pathway analysis, and survival rate analysis

Total RNA was respectively isolated from U87-MG cells with or without 200 ng/ml IGF-1 treatment for 48 h. MiRNA expression profiles were performed using Human microRNA OneArray® vers. 5.1 (Phalanx Biotech Group, Hsinchu, Taiwan). All experiments including complementary (c)RNA amplification, hybridization, image scanning with an Axon 4000 scanner (Molecular Devices, Sunnyvale, CA, USA), and statistical analysis with Genepix software (Molecular Devices) were conducted by the Phalanx Biotech Group (Hsinchu, Taiwan). The log2 (ratio) was calculated by the pair-wise combination and error weighted average. All the array data were uploaded to Gene Expression Omnibus (GEO) database as accession number GSE140297. Adjusted (adj.) *p* values were calculated by the Benjamini-Hochberg multiple testing correction. Significant differentially expressed (DE) gene lists filtered for adj. *p* values (DE) of <0.05 and a |log2 (ratio)| of ≥0.58 (±1.5 multiples of change) cutoff were applied for further analysis. A hierarchical clustering analysis and heat map were carried out using

CIMminer [21]. Target genes of miRNAs were predicted by TargetScan 6.2 [22]. Microarray data from the GEO dataset were analyzed by GEO2R.

## RNA isolation and quantitative real-time reverse-transcription polymerase chain reaction (RT-qPCR)

Total RNA from cultured cells was extracted using Trizol® according to the manufacturer's instructions. RNA quality was checked using A260/A280 readings. cDNA was synthesized from 1 μg of total RNA using a random primer and the MultiScribe (tm) Reverse Transcriptase Kit. cDNA was diluted 1:30 with PCR-grade water and then stored at -20˚C. To detect miRNA levels, cDNA was synthesized with a TaqMan® Advanced miRNA cDNA Synthesis Kit (Applied Biosystems). TaqMan® Advanced miRNA assays were used to detect miR-513a-5p and miR-191-5p. To quantify miRNA expression levels, miR-191-5p was used as an internal control. To detect primary (pri)-miR-513a levels, TaqMan® pri-miR-miR-513a assays were used. Specific primers for human NEDD4L and GAPDH for the real-time qPCR are listed in S1 Table. Gene expression levels were quantified with the Applied Biosystems StepOnePlus™ System (Thermo Fisher Scientific) with pre-optimized conditions. Each PCR was performed in triplicate using 5 μl of 2x SYBR Green PCR Master Mix, 0.2 μl of primer sets, 1 μl cDNA, and 3.6 μl nucleotide-free $H_2O$ to yield 10 μl per reaction. Expression rates were calculated as the normalized $C_T$ difference between the control and sample after adjusting for the amplification efficiency relative to the expression level of GAPDH.

## Construction of the NEDD4L 3'UTR reporter plasmid and mutagenesis

The PCR was performed using sets of primers specific for the NEDD4L 3'UTR, of which the forward primer was XhoI-site-linked and the reverse primer was XbaI-site-linked. PCR primers are listed in S1 Table. U87 MG genomic DNA was used as a template. The 535-base pair (bp) PCR product was digested with XhoI and XbaI, and cloned downstream of the luciferase gene in the pMIRGLO-REPORT luciferase vector (Promega). This vector was sequenced and named pMIRGLO-NEDD4L-3'UTR. Site-directed mutagenesis of the miR-513a-5p target site in the NEDD4L 3'UTR was carried out using an overlapping PCR, and the vector was named pMIRGLO-NEDD4L-3'UTR-mutant. For reporter assays, cells were transiently transfected with the wild-type or mutant reporter plasmids, and miR-513a-5p plasmids using Lipofectamine 3000 (Invitrogen). The reporter assay was performed at 24 h post-transfection using the Luciferase Assay System (Promega). The dual Renilla luciferase value was used as an internal control.

## Construction of full-length NEDD4L cDNA and miR-513a-5p-overexpressing plasmids

Full-length NEDD4L cDNA (NM_001144966) without the 3'UTR was generated by PCR amplification using primers listed in S1 Table. To construct miR-513a-5p-overexpressing plasmids, the 300-bp length of the miR-513a-5p gene was generated by PCR amplification using primers listed in S1 Table. The following thermal profile was used for the PCR amplification of 500 ng of cDNA on a GeneAmp PCR system 9700 (Applied Biosystems): an initial denaturation step at 95˚C for 5 min, followed by 40 cycles of 94˚C for 1 min, 58˚C for 1 min, and 72˚C for 1 min, with a final extension at 72˚C for 10 min. PCR products were analyzed by agarose gel electrophoresis. All PCR products were cloned into pGEM-T Easy (Promega) and sequenced. After *Bam*HI/*Eco*RI digestion, NEDD4L and miR-513a-5p cDNA was cloned into

pCDH vectors (System Biosciences, Palo Alto, CA, USA) to form constructs named pCDH-NEDD4L and pCDH-miR-513a-5p.

## TOPFlash luciferase reporter assays

A Luciferase assay was conducted by transfecting 1 μg TOPFLASH (Firfly) and 4 ng TK-Nluc (NanoLuc) plasmids into glioma cells. After 4 h incubation, cells were treated with 200ng/ml IGF-1 overnight. Cell lysates were harvested and added with different substrate to measure Firfly and NanoLuc luminesces subsequently. Then, the ratio between Firfly and NanoLuc was calculated and represent β-catenin mediated transcriptional activity.

## Nuclear protein extraction

After harvesting cells through trypsinization, cell lysates were resuspended in 2 ml of pre-chilled PBS, 2 ml of nuclear isolation buffer (1.28 M sucrose, 40 mM Tris-HCl pH 7.5, 20 mM $MgCl_2$, and 4% Triton X-100), and 6 ml of sterilized water. Then, cells were incubated on ice for 20 min. Nuclei were pelleted by centrifugation at 2,500 *g* for 15 min, washed once with 1 ml of nuclear isolation buffer, and re-suspended in RIPA. The nuclear pellets were collected for immunoblotting assays.

## Oxidative DNA damage measurement

DNA damage levels (8-OHdG) were measured by using OxiSelect™ Oxidative DNA Damage ELISA Kits (CELL BIOLABS, INC.; San Diego, CA, USA) according to the manufacturer's instructions. DNA from treated cells was extracted by a commercial DNA Extraction kit. Fifty microliter of unknown sample or 8-OHdG standard was respectively added to the wells of the 8-OHdG Conjugate coated plate. After incubation for 10 minutes at room temperature, 50 μL of the diluted anti-8-OHdG antibody was added to each well for another 1 h incubation. After washing 3 times with 250 μL 1X Wash Buffer per well, 100 μL of the diluted Secondary Antibody-Enzyme Conjugate was added to all wells for 1 h incubation. After washing, 100 μL of Substrate Solution was added to each well and incubated for 1h. After adding 100 μL of Stop Solution into each well, the absorbance was measured by using a spectrophotometer with 450 nm wave length.

## Statistical analysis

All data are presented as the mean ± standard deviation (SD). Significant differences among groups were determined using an unpaired Student's *t*-test. A value of $p < 0.05$ was taken as an indication of statistical significance. All figures shown in this article were obtained from at least three independent experiments with similar results.

# Results

## IGF-1 attenuates glioma cell sensitivity to TMZ via Wnt/β-catenin signaling

To test the effects of IGF-1 on glioma U87-MG cell growth, different doses of IGF-1 were used for the indicated times (Fig 1A and 1B). Surprisingly, we found that IGF-1 exhibited mild effects on U87-MG cell growth. IGF-1 at 200 ng/ml only increased the glioma cell growth ratio by 13% using MTT assays and 21% using cell counting assays compared to the control after 48 h of treatment. To explore the molecular functions of IGF-1 on glioma cell progression, we conducted a literature search. Since Vanamala et al. [23] reported that IGF-1 could activate the WNT/β-catenin pathway in colon cancer, we then tested the effects of IGF-1 on activating the

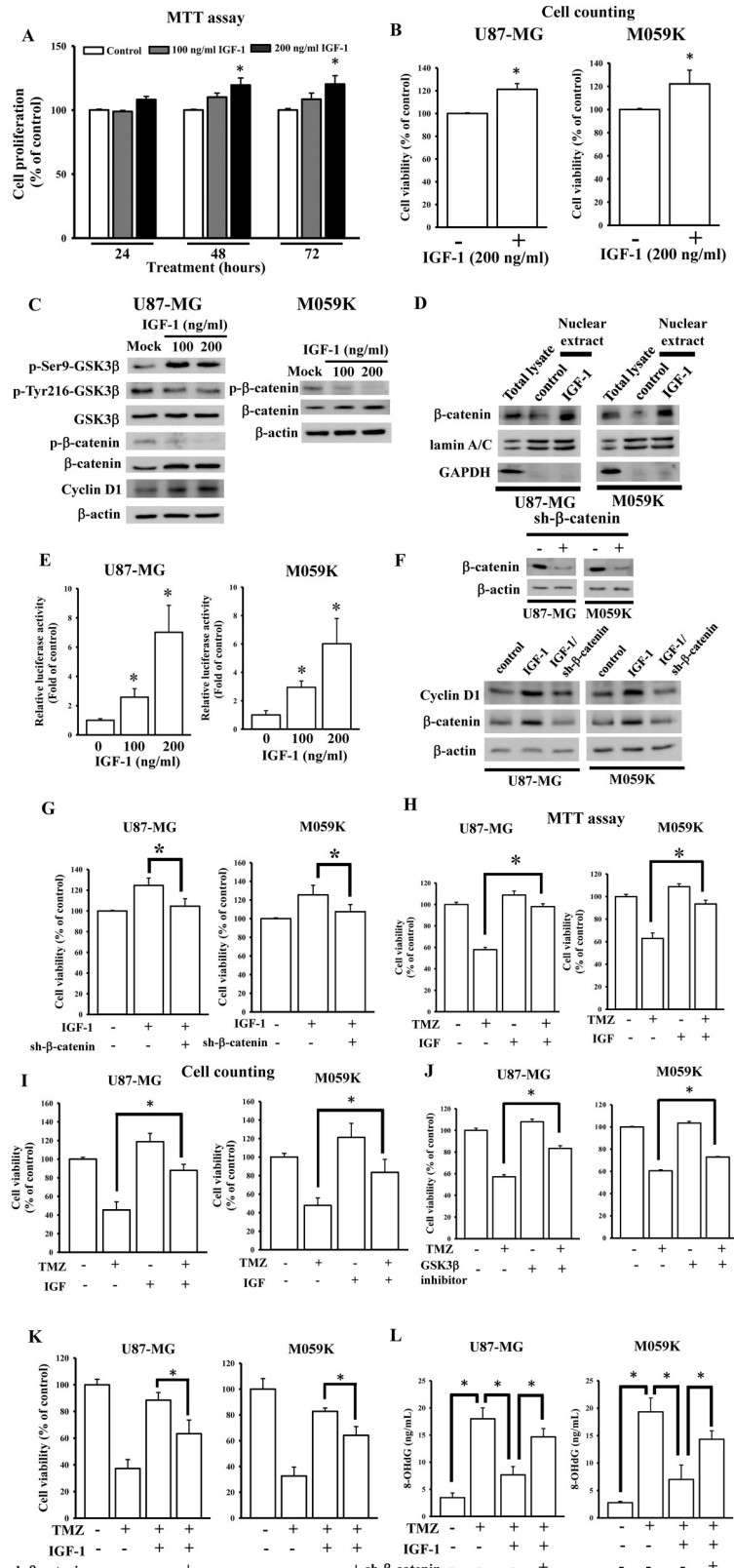

**Fig 1. The insulin-like growth factor (IGF)-1-activated WNT/β-catenin pathway influences glioma cell sensitivity to temozolomide (TMZ).** (A and B) Effects of IGF-1 on cell proliferation. U87 MG cells were treated with different

doses of IGF-1 for the indicated times or indicated doses of IGF-1 for 48 h. The cell proliferation ratio was respectively measured by an MTT assay and cell counting assay with Trypan blue stain. Data are the mean ± SD of three independent experiments. * $p < 0.05$. (C) Validation of IGF-1's effect on activation of the WNT/β-catenin pathway. After cells were treated with indicated doses of IGF-1 for 48 h, phosphorylation changes and total levels of glycogen synthase kinase 3β (GSK3β), β-catenin, and cyclin D1 expression levels were detected by immunoblotting assays. (D) The effects of IGF-1 on β-catenin nuclear accumulation. (E) IGF-1 enhances β-catenin-mediated transcriptional activity. The detail protocol for nuclear protein extraction and TOPFlash luciferase reporter assays were showed in method sections. Nuclear β-catenin, lamin A/C, and GAPDH levels were detected by immunoblotting assays. Data are the mean ± SD of three independent experiments. * $p < 0.05$. (F and G) Knockdown effects of β-catenin on IGF-1-induced cyclin D1 levels and cell proliferation. β-catenin and cyclin D1 expression were detected by immunoblotting assays. Cell viability was measured by using cell counting assays. Data are the mean ± SD of three independent experiments. * $p < 0.05$. (H and I) Effects of IGF-1 on reducing TMZ cytotoxicity. The cell viability was respectively measured by an MTT assay and cell counting assay with Trypan blue stain. Data are the mean ± SD of three independent experiments. * $p < 0.05$. Treatment with GSK3β inhibitor VIII (J) or β-catenin shRNAs (K) antagonized IGF-1-reduced TMZ cytotoxicity and DNA damage (L). After cells were transfected with 1 µg β-catenin shRNAs for 24 h or 1 µM of GSK3β inhibitor VIII for 1h, 200 µM TMZ or combined with 200 ng/ml IGF-1 was added for 48 h. Cell viability was measured by an MTT assay or cell counting assay. DNA damage levels were measured by using OxiSelect™ Oxidative DNA Damage ELISA Kits according to the manufacturer's instructions. Data are the mean ± SD of three independent experiments. * $p < 0.05$.

WNT/β-catenin pathway in glioma U87-MG cells. As shown in Fig 1C, IGF-1 treatment enhanced phosphorylation levels in Ser9 of GSK3β (inactive form) and reduced Tyr216 phosphorylated levels (active form), resulting in inhibition of GSK3β activation. Then, these reduced phosphorylation levels of β-catenin led to increases in its endogenous level and downstream regulator expressions such as cyclin D1, suggesting that IGF-1 activated the WNT/β-catenin pathway in glioma U87-MG cells. Similarly, IGF-1-activated β-catenin was also observed in M059K cells (Fig 1C, right panel). To further test the effects of IGF-1 on β-catenin activation, we measured β-catenin nuclear accumulation levels and conducted a TOPFlash luciferase reporter assay whose promoter contains TCF/LEF binding sites for β-catenin binding. We found that IGF-1 treatment significantly enhanced β-catenin nuclear accumulation (Fig 1D) and increased luciferase activity (Fig 1E). Furthermore, knockdown of endogenous β-catenin levels significantly attenuated IGF-1-induced cyclin D1 (Fig 1F) and cell growth (Fig 1G), suggesting that IGF-1 induced β-catenin signaling activation in glioma cells.

A previous study reported that the WNT/β-catenin pathway plays an important role in enhancing chemoresistance mechanisms of glioma cells [24]. We then investigated the effects of IGF-1 on U87-MG cell sensitivity to TMZ. As shown in Fig 1H and 1I, respectively using MTT assays and cell counting assays, treatment with 200 ng/ml IGF-1 significantly attenuated TMZ cytotoxicity in U87-MG cells. Similar results were also found in M059K cells treated with IGF-1 and TMZ (Fig 1H and 1I, right panel). Furthermore, after treating U87-MG cells with the GSK3β inhibitor VIII, we found that inhibition of GSK3β activity significantly attenuated TMZ cytotoxicity (Fig 1J). Knockdown of endogenous β-catenin levels also reduced IGF-1-insensitized TMZ cytotoxicity (Fig 1K) and DNA damage (Fig 1L), suggesting that IGF-1 reduces glioma cell sensitivity to TMZ treatment via activating the WNT/β-catenin pathway.

## Elevated miR-513a-5p levels were found in IGF-1-mediated miRNA signatures and GBM miRNA profiles

To explore whether miRNAs participate in IGF-1-mediated glioma cell sensitivity to TMZ treatment, we conducted an miRNA microarray analysis using cells treated with 200 ng/ml IGF-1 for 48 h. We selected significant differentially expressed miRNAs with a log2 (ratio) of $|\geq 0.58|$ and an adj. $p$ value (DE) of $<0.05$. We found that IGF-1 treatment enhanced 93 miRNA expressions and downregulated 148 miRNA levels (Fig 2A and 2B). From the miRNA microarray results (S1 File), miR-513a-5p was the most significantly upregulated miRNA in

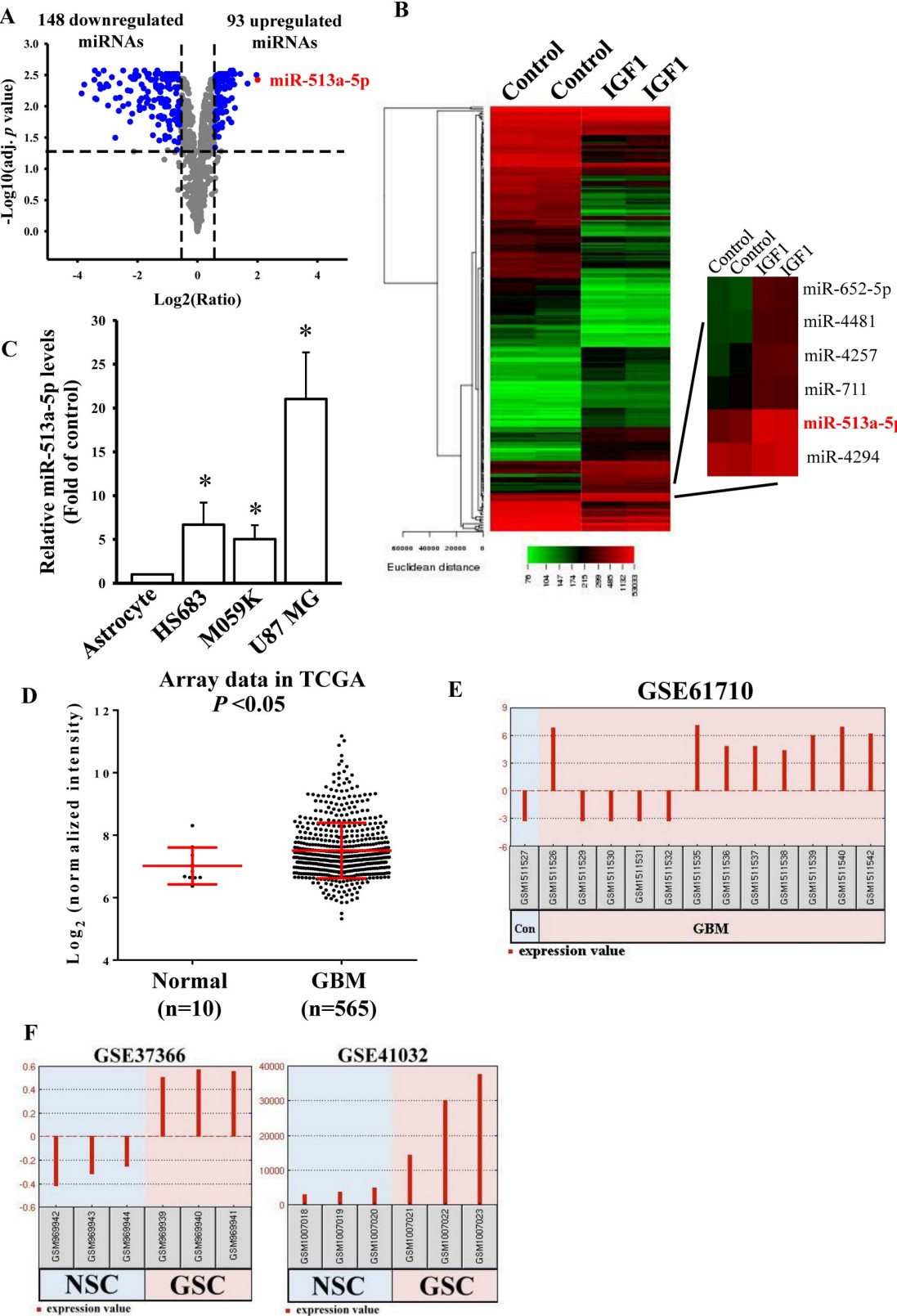

**Fig 2. Significant upregulation of microRNA (miR)-513a-5p in insulin-like growth factor (IGF)-1-treated glioma cells.**
Volcano plot (A) and heat map of hierarchical gene clustering (B) demonstrating IGF-1-regulated microRNA profiles. After U87

MG cells were treated with 200 ng/ml IGF-1 for 48 h, total RNAs were extracted for a microarray analysis. For all array analyses, an adjusted (adj.) p value of <0.05 and a |log2 (ratio)| of ≥0.58 (±1.5 multiples of change) cutoff were applied. Volcano plots show the multiples of change (log$_2$ ratio) and probability (-log$_{10}$ adj. p values) of individual microRNAs from the microarray assay. The red dot shows the miR-513a-5p location. The heat map depicts the 241 microRNAs differentially expressed between IGF-1 subsets. A color was assigned to each microRNA based on its relative expression level across samples. (C) Detection of endogenous miR-513a-5p levels in human normal astrocytes and three different glioma cell lines. Endogenous miR-513a-5p levels were measured by a real-time PCR. The miR-191-5p level was used as an internal control. Data are the mean ± SD of three independent experiments. * p<0.05. Increased miR-513a levels in GBM patients and glioma stem cells (GSCs) compared to normal controls from TCGA miRNA microarray data (D), and GEO DataSets GSE61710 (E), GSE37366, and GSE41032 (F). (E and F) miR-513a-5p expression changes were analyzed by GEO2R software.

IGF-1-treated U87 MG cells (S2 and S3 Tables). To investigate expression levels of miR-513a-5p in glioma cells and patients, we first compared endogenous levels of miR-513a-5p in four cell lines including primary astrocytes, and HS683, M059K, and U87-MG glioma cells. Higher expression levels of miR-513a-5p were observed in glioma cells compared to primary astrocytes (Fig 2C). In addition, by analyzing the miRNA array data in TCGA database, higher miR-513a expression levels were found in GBM patients compared to normal controls (Fig 2D). Furthermore, by analyzing GSE61710 array data in the GEO dataset with GEO2R software, we found that eight of 12 GBM patients had elevated miR-513a-5p levels compared to the normal brain control (Fig 2E). Moreover, in GSE37366 and GSE41032 array data (Fig 2F), we also observed that glioma stem cells (GSCs) possessed higher miR-513a-5p levels than did neural stem cells (NSCs). As a consequence, we concluded that higher miR-513a-5p levels participate in the GBM process.

## IGF-1 upregulated *miR-513a-2* but not *miR-513a-1* expression via PI3K pathways

To reconfirm the array results, we measured the effects of IGF-1 on miR-513a-5p expression levels. As shown in Fig 3A, IGF-1 significantly and dose-dependently enhanced miR-513a-5p levels in both U87-MG and M059K cells. miR-513a-5p is an intergenic miRNA and comes from two different gene loci, *miR-513a-1* and *miR-513a-2* in the X chromosome. To investigate which of these two loci is mainly regulated by IGF-1, we first measured changes in expression of the two primary (pri) forms of miR-513a after IGF-1 treatment. As shown in Fig 3B, IGF-1 significantly enhanced pri-miR-513a-2 expression levels and only had a mild effect on pri-miR-513a-1 expression. Since PI3K- and ERK-mediated signaling are two major downstream pathways in IGF-1 signaling [25], we then investigated which signaling pathway played a critical role in IGF-1-stimulated miR-513a-5p expression. After U87-MG cells were respectively co-treated with IGF-1 and LY294002 (a PI3K inhibitor) or U0126 (an ERK inhibitor) for 48 h, relative miR-513a-5p and pri-miR-513a-2 levels were measured. As shown in Fig 3C and 3D, only LY294002 showed inhibitory effects on IGF-1-enhanced miR-513a-5p and pri-miR-513a-2 levels, but U0126 did not. Taken together, IGF-1 transcriptionally enhances miR-513a-5p expression levels via PI3K signaling.

## Reduced NEDD4L levels in GBM patients and glioma cell lines

A previous study reported that NEDD4L is a negative regulator of the Wnt/β-catenin pathway in colon cancer [17]. However, its role in GBM is still unclear. By comparing expression levels of NEDD4L in GBM patients with normal controls using microarray and RNA sequencing (RNA Seq) data from TCGA database, we found that NEDD4L levels were significantly reduced in GBM patients compared to those in control groups (Fig 4A and 4B). Furthermore, we next compared endogenous NEDD4L levels in normal astrocytes and three different glioma

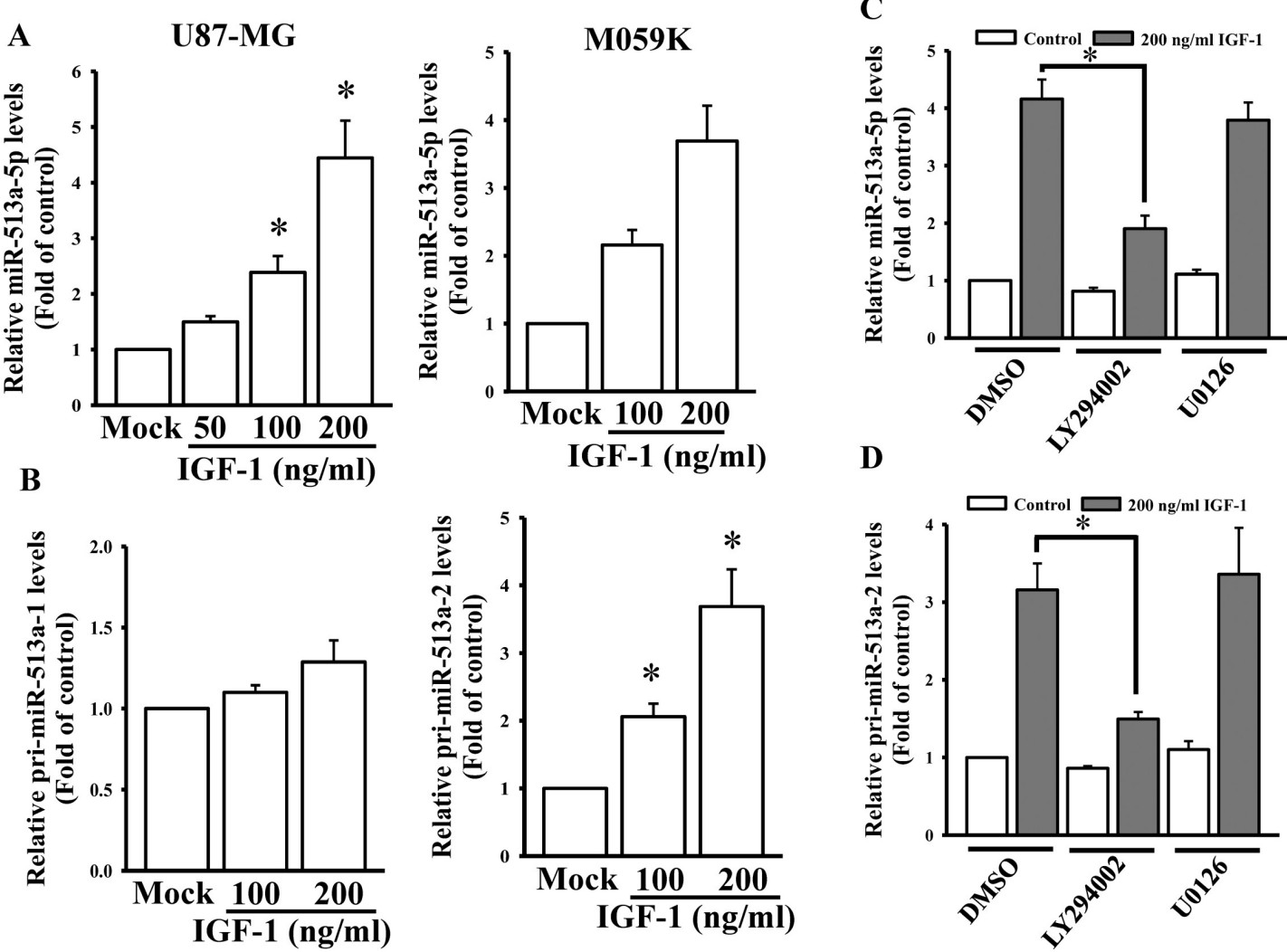

**Fig 3. Insulin-like growth factor (IGF)-1 upregulates miR-513a-5p expression from the *miR-513a-2* gene via the phosphoinositide 3-kinase (PI3K) pathway.** (A) Dose-dependent effects of IGF-1 on miR-513a-5p expression levels in U87-MG and M059K cells. (B) Effects of IGF-1 on primary (pri)-miR-513a-1 and pri-miR-513a-2 levels. The PI3K inhibitor, LY294002, but not the mitogen-activated protein kinase (MAPK) kinase inhibitor, U0126, attenuated IGF-1-stimulated miR-513a-5p (C) and pri-miR-513a-2 (D) levels. After cells were treated with the indicated doses of IGF-1 for 48 h or 200 ng/ml IGF-1 treatment for 48 h and pretreatment with 5 μM LY294002 and 10 μM U0126 for 1 h, endogenous miR-513a-5p, pri-miR-513a-1, and pri-miR-513a-2 levels were respectively measured by a real-time PCR. miR-191-5p and GAPDH levels were respectively used as internal controls for the mature and primary miR-513a-5p levels. Data are the mean ± SD of three independent experiments. * $p < 0.05$.

cell lines. As shown in Fig 4C and 4D, both the protein and mRNA levels of NEDD4L were lower in glioma cells than in primary astrocytes. To explore the role of NEDD4L in regulating glioma cell death, we first cloned NEDD4L cDNA into pCDH vectors (Fig 4E). After NEDD4L overexpression, the viability of both U87-MG and M059K cells was significantly and dose-dependently decreased (Fig 4F), suggesting that NEDD4L acts as a tumor suppressor in glioma progression.

## Identification of NEDD4L as a direct target gene of miR-513a-5p

To investigate the correlation between miR-513a and NEDD4L levels in GBM, we conducted a Pearson's correlation analysis with TCGA microarray data. No significant correlation was

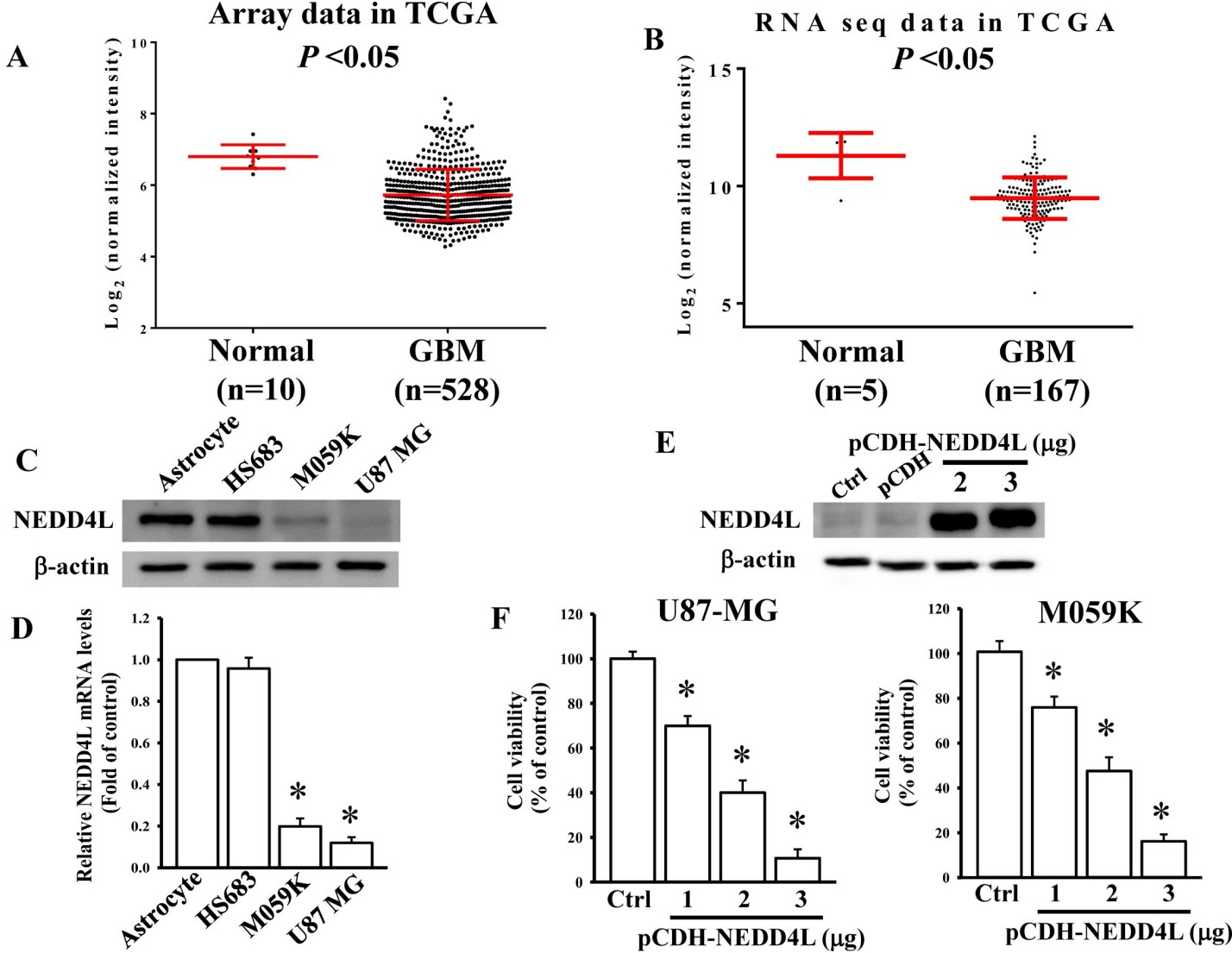

**Fig 4. Overexpression of NEDD4L levels induces glioma cell death.** (A and B) Relative NEDD4L expression levels in glioblastoma multiforme (GBM) patients compared to normal control groups from the microarray and RNA sequencing (RNA Seq) data of TCGA database. Detection of endogenous NEDD4L protein (C) and mRNA (D) levels in human normal astrocytes and three different glioma cell lines. Endogenous NEDD4L protein and mRNA levels were respectively measured by an immunoblotting assay and a real-time PCR. β-Actin and GAPDH levels were used as internal controls. Data are the mean ± SD of three independent experiments. * $p<0.05$. (E) Detection of NEDD4L levels in NEDD4L-overexpressing cells. (F) Overexpression of NEDD4L reduced glioma cell viability. After cells were transfected with the indicated doses of NEDD4L cDNA-containing plasmids for 24 h, NEDD4L protein levels were measured by immunoblotting assays. Cell viability was measured by an MTT assay. Data are the mean ± SD of three independent experiments. * $p<0.05$.

found between miR-513a and NEDD4L levels in all GBM patients (S1A Fig). When further dividing patients into two groups using the median cutoff for miR-513a expression levels, we found that a significant inverse correlation between miR-513a and NEDD4L levels existed in patients highly expressing miR-513a (Fig 5A), but not in the group expressing miR-513a at a lower level (S1B Fig). All of the data suggested that only higher miR-513a levels could influence endogenous NEDD4L expression. Using the TargetScan 6.2 prediction [22], NEDD4L was predicted to be a putative target gene of miR-513a-5p (Fig 5B). To further confirm that NEDD4L is the target gene of miR-513a-5p, the 3'UTR of the NEDD4L gene containing a miR-513a-5p-binding site was cloned into the pmiRGlo-reporter plasmid to conduct 3'UTR

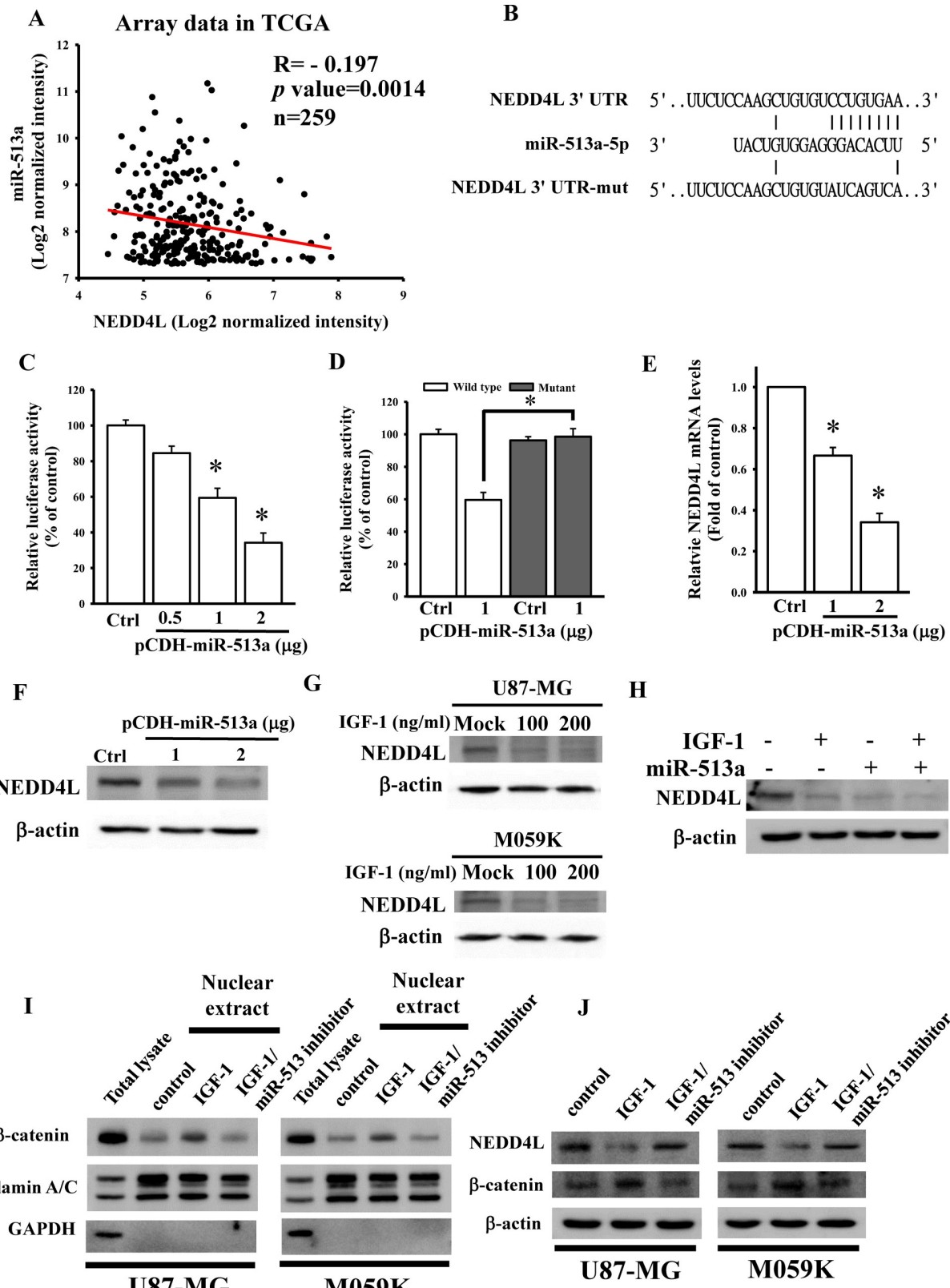

**Fig 5. Insulin-like growth factor (IGF)-1 reduced NEDD4L expression through miR-513a-5p targeting.** (A) The inverse correlation between miR-513a and NEDD4L expression levels in TCGA microarray data. Using the median cutoff for miR-513a expression levels,

patients were divided into two groups. A correlation was found in the group highly expressing miR-513a ($n$ = 259) by Pearson's correlation analysis. (B) Schematic diagram of potential miR-513a-5p-targeted sites in the NEDD4L 3'-untranslated region (UTR). (C and D) Effects of miR-513a-5p on NEDD4L 3'-UTR luciferase activity. To test for miR-513a-5p's effect, different doses of the miR-513a-5p plasmid were co-transfected with 500 ng of the pmiRGlo-NEDD4L 3'-UTR or mutant 3'-UTR. Luciferase activity was measured in these cells 24 h after transfection. Effects of miR-513a-5p overexpression on NEDD4L mRNA (E) and protein (F) expressions. After cells were transfected with the indicated dose of the miR-513a-5p plasmid for 24 h, relative mRNA and protein levels of NEDD4L were analyzed using a real-time PCR and an immunoblotting assay. (G) Dose-dependent effects of IGF-1 on reducing NEDD4L levels. (H) miR-513a-5p is involved in the IGF-1-reduced NEDD4L expression mechanism. After cells were treated with indicated doses of IGF-1 for 48 h or transfected with 1 μg of the miR-513a-5p plasmid combined with IGF-1 treatment for 48 h, the NEDD4L protein level was analyzed by an immunoblotting assay. Data are the mean ± SD of three experiments. * $p$<0.05. Inhibitory effects of miR-513a-5p on IGF-1-induced β-catenin nuclear accumulation (I) and -reduced NEDD4L levels (J). The protocol for nuclear protein extraction was showed in the method section. The β-catenin, lamni A/C, GAPDH, and NEDD4L protein levels were analyzed by immunoblotting assays.

reporter assays. As shown in Fig 5C, different concentrations of miR-513a-5p-overexpressing plasmids significantly decreased luciferase activities of NEDD4L. To further validate that NEDD4L is inhibited by miR-513a-5p via binding to the 3'UTR, five nucleotides located in the critical binding region of the 3'UTR of the NEDD4L gene were mutated by mutagenesis (Fig 5B). As shown in Fig 5D, miR-513a-5p had no effect on luciferase activity after mutating the miR-513a-5p-targeted site. We also directly tested the effect of miR-513a-5p on NEDD4L expression and found that transient transfection of miR-513a-5p into U87 MG cells significantly and dose-dependently decreased mRNA and protein levels of NEDD4L as measured by real-time qPCR assays (Fig 5E) and an immunoblotting analysis (Fig 5F). To explore the effects of IGF-1 stimulation on NEDD4L levels, U87-MG and M059K cells were respectively treated with the indicated doses of IGF-1. As shown in Fig 5G, IGF-1 treatment significantly reduced endogenous NEDD4L levels in glioma cells. To further identify if miR-513a-5p is really involved in IGF-1-repressed NEDD4L levels, we measured NEDD4L protein levels after transfection of miR-513a-5p-overexpressing plasmids combined with IGF-1 treatment. As shown in Fig 5H, overexpression of miR-513a-5p expression significantly enhanced IGF-1-reduced NEDD4L protein levels. In contrast, inhibition of miR-513a-5p expression significantly reduced IGF-1-enhanced β-catenin nuclear accumulations (Fig 5I) and -decreased NEDD4L expression (Fig 5J). Taken together, IGF-1-mediated miR-513a-5p upregulation significantly inhibited NEDD4L expressions in glioma cells.

## miR-513a-5p-inhibited NEDD4L expression influenced IGF-1-mediated WNT/β-catenin signaling in attenuating glioma cell sensitivity to TMZ

To investigate the effects of miR-513a-5p on the NEDD4L-repressed WNT/β-catenin pathway, different doses of miR-513a-5p-overexpressing plasmids were transfected into U87-MG cells. miR-513a-5p significantly reduced NEDD4L levels and enhanced β-catenin and cyclin D1 expressions, but did not influence phosphorylated or total levels of GSK-3β (Fig 6A). Furthermore, overexpression of miR-513a-5p showed greater promoting effects on β-catenin signaling in IGF-1-treated U87-MG cells (Fig 6B). In contrast, overexpression of NEDD4L significantly attenuated IGF-1-enhanced β-catenin signaling (Fig 6C). Finally, we tested the effects of miR-513a-5p and NEDD4L on IGF-1-attenuated U87-MG cell sensitivity to TMZ treatment. As shown in Fig 6D and 6E, overexpression of miR-513a-5p or NEDD4L significantly influenced IGF-1-mediated U87-MG cell sensitivity to TMZ treatment. In contrast, knockdown of miR-513a-5p reduced IGF-1-insensitized TMZ cytotoxicity (Fig 6F). Taken together, IGF-1-upregulated miR-513a-5p expression decreased U87-MG cell sensitivity to TMZ via targeting NEDD4L-inhibited WNT/β-catenin signaling.

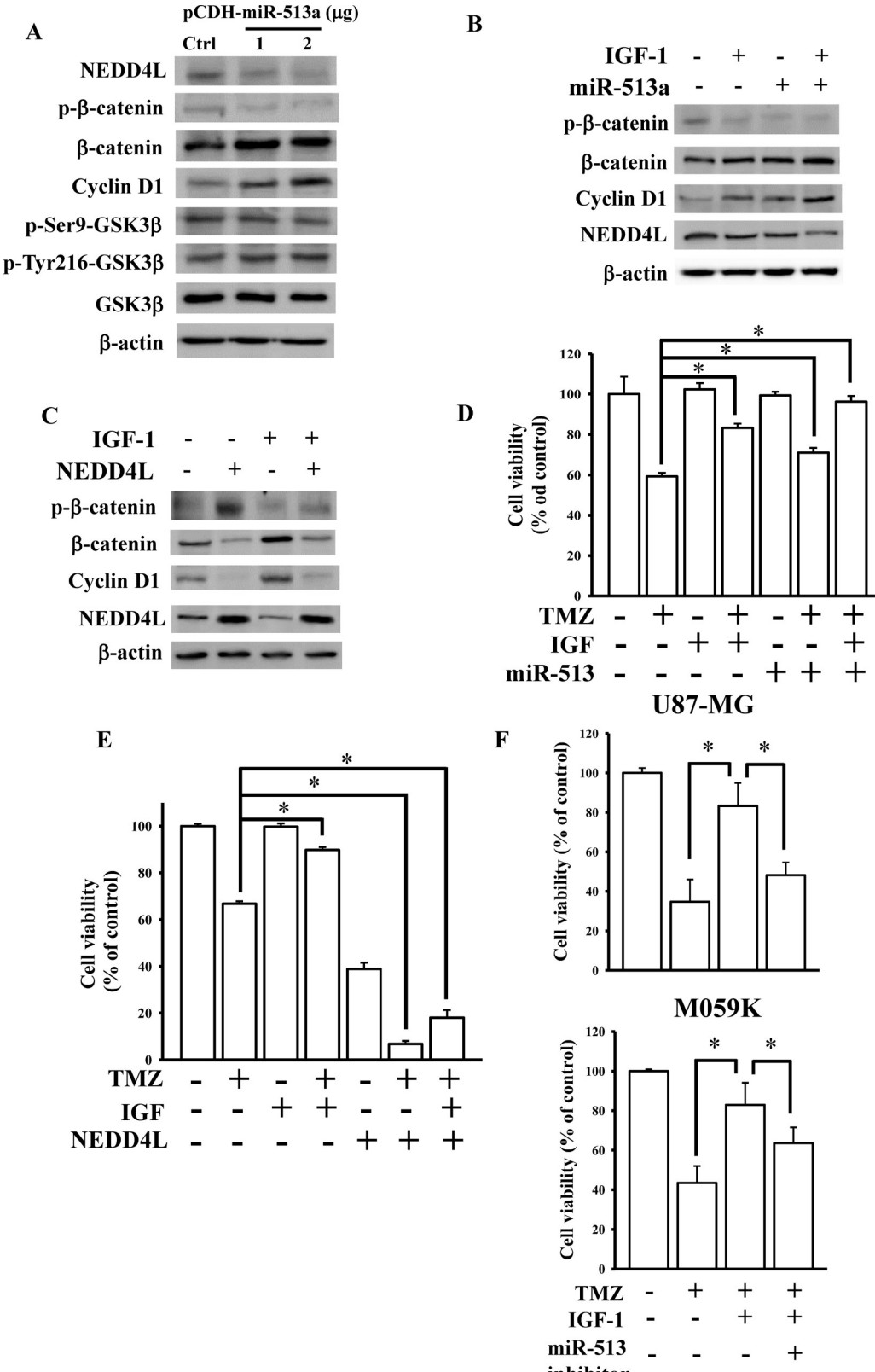

**Fig 6. miR-513a-5p inhibited NEDD4L signaling in insulin-like growth factor (IGF)-1-desensitized glioma cells to temozolomide (TMZ).** (A) Effects of the overexpression of miR-513a-5p on downstream protein expressions of

NEDD4L-inhibited WNT/β-catenin signaling. (B) Overexpression of miR-513a-5p enhanced the IGF-1-activated WNT/β-catenin pathway. (C) Overexpression of NEDD4L repressed the IGF-1-activated WNT/β-catenin pathway. After U87-MG cells were transfected with the indicated doses of the miR-513a-5p plasmid for 24 h, 1 μg of the miR-513a-5p plasmid combined with 200 ng/ml IGF-1 treatment for 48 h, or 2 μg NEDD4L plasmids combined with 200 ng/ml IGF-1 treatment for 48 h, WNT/β-catenin pathway downstream protein expressions were measured by an immunoblotting assay. Effects of miR-513a-5p (D) and NEDD4L (E) on IGF-1-reduced glioma sensitivity to TMZ. After U87-MG cells were transfected with 1 μg of the miR-513a-5p plasmid or 2 μg of the NEDD4L plasmid combined with 200 ng/ml IGF-1 and 200 μM TMZ treatment for 48 h, cell viability was measured by an MTT assay. Data are the mean ± SD of three independent experiments. * $p < 0.05$. (F) Knockdown effects of miR-513a-5p on IGF-1-insensitized TMZ cytotoxicity. Cell viability was measured by a cell counting assay. Data are the mean ± SD of three independent experiments. * $p < 0.05$.

## Discussion

IGF-1/IGF-1R signaling was identified as protecting against GBM cell apoptosis [26]. The IGF-1R is also considered an anticancer target and a therapeutic strategy to repress GBM progression [27, 28]. In addition, inhibition of IGF-1R activity using an ATP-competitive IGF-1R/IR inhibitor induced dramatic apoptosis in TMZ-resistant cells [29]. However, no studies have reported the role of the IGF-1 axis in the formation of TMZ resistance of GBM. Herein, we demonstrated a distinct role of IGF-1 signaling in influencing TMZ drug sensitivity. We found that IGF-1 treatment significantly reduced TMZ's cytotoxic responses toward GBM cells via WNT/β-catenin signaling. Using miRNA array analyses, we identified that miR-513a-5p is upregulated in IGF-1-treated GBM cells via activating PI3K signaling. Overexpression of miR-513a-5p significantly improved IGF-1 treatment by desensitizing the TMZ response. NEDD4L, an E3 ubiquitin protein ligase involved in inhibiting WNT signaling, was also identified as a direct target gene of miR-513a-5p. Finally, we concluded that IGF-1 activated WNT/β-catenin signaling through miR-513a-5p-inhibited NEDD4L expression, consequently resulting in reduced TMZ sensitivity toward GBM cells.

WNT/β-catenin signaling was shown to play a critical role in GBM by enhancing resistance to chemotherapy, especially TMZ [30, 31]. Therefore, several inhibitors which target members of the WNT signaling pathway have gradually been developed for GBM therapy [32]. Additionally, WNT/β-catenin signaling was also reported to be involved in the IGF-1 axis [33, 34]. However, a gap still exists between IGF-1/IGF-1R and WNT/β-catenin signaling. In the present study, we identified that IGF-1 treatment significantly reduced NEDD4L expression levels, resulting in activation of WNT/β-catenin signaling. Furthermore, PI3K signaling-activated miR-513a-5p also participated in IGF-1-reduced NEDD4L levels by directly targeting the NEDD4L 3'UTR. Consequently, although GSK-3β plays a critical role in activating WNT/β-catenin signaling [35], our results suggested that miR-513a-5p-reduced NEDD4L signaling is important for persistent activation of IGF-1-enhanced WNT/β-catenin signaling.

Recently, several studies discovered that miRNAs are involve in modulating drug resistance. For example, ectopic overexpression of miR-7-5p [36] and miR-181b [37] sensitized GBM cells to TMZ treatment by respectively targeting Yin Yang 1 and the epidermal growth factor receptor. In addition, miR-513a-5p was also recognized as a potential onco-miR and risk factor in breast cancer [38]. However, until now, no study has reported the physiological roles of miR-513a-5p in GBM progression and chemoresistance. Herein, we found that elevated miR-513a-5p levels existed in GBM cells and patients compared to control groups. Significant upregulation of miR-513a-5p was also observed in IGF-1-treated GBM cells. By targeting NEDD4L, miR-513a-5p reduced the TMZ treatment response of GBM cells, suggesting that IGF-1-induced miR-513a-5p signaling participates in affecting the chemosensitivity of TMZ toward GBM. miR-513a-5p may play a crucial role in enhancing GBM progression.

NEDD4L (NEDD4-2 and NEDD4.2), which belongs to the HECT E3 ubiquitin ligase family [39], plays a key role in modulating protein ubiquitination. A previous study reported that

downregulation of NEDD4L is observed in human gliomas, and patients with lower NEDD4L levels had poor prognoses [19]. However, its function in glioma progression is still unclear. As with previous findings [17, 40], we identified that NEDD4L was downregulated in GBM and inhibited canonical WNT/β-catenin signaling. Furthermore, we also found that IGF-1-inhibited NEDD4L signaling desensitized GBM cells to TMZ treatment. In addition, investigating the molecular mechanisms would also be helpful for exploring the roles of NEDD4L in disease progression. Previous studies suggested that NEDD4L levels were modulated through post-transcriptional regulation. For example, the AU-rich element RNA-binding protein (AUF1) increases NEDD4L levels via interacting with the 3'UTR of NEDD4L mRNA [41]. In contrast, reduced NEDD4L expression was induced by several distinct miRNAs such as miR-93 [42], miR-1 [43], and the miR-106b-25 cluster [44]. In the present study, NEDD4L was identified as a direct target gene of miR-513a-5p that was enhanced by IGF-1 treatment. miR-513a-5p-repressed NEDD4L signaling participated in IGF-1's activation of WNT/β-catenin signaling, and consequently in reducing TMZ cytotoxicity. In conclusion, our results provide convincing evidence of IGF-1's effects on sensitizing GBM cells to TMZ through activating miR-513a-5p/ NEDD4L/WNT/β-catenin signaling. Our findings may provide new avenues for developing therapeutic targets and drugs in TMZ-insensitive GBM disease.

## Supporting information

**S1 Fig. The correlation between miR-513a and NEDD4L expression levels in TCGA micro-array data.** (A) Total patients (n = 519). (B) The group with lower miR-513a expression levels (n = 260). By using the median cutoff for miR-513a expression levels, the patients were divided into two groups. The correlation was calculated by Pearson's correlation analysis.
(PDF)

**S2 Fig. Raw data of immunoblotting assays.**
(PDF)

**S1 Table. Primer list.**
(PDF)

**S2 Table. List of IGF-1-upregulated miRNAs in U87 MG cells (log2 (Ratio) ≧ 0.58; adj. *p* value ≦0.05).**
(PDF)

**S3 Table. List of IGF-1-downregulated miRNAs in U87 MG cells (log2 (Ratio) ≦-0.58; adj. *p* value ≦0.05).**
(PDF)

**S1 File. Normalized microarray data.**
(XLSX)

## Author Contributions

**Conceptualization:** Ku-Chung Chen, Chia-Hsiung Cheng, Chin-Cheng Lee.

**Data curation:** Ku-Chung Chen, Peng-Hsu Chen, Kuo-Hao Ho.

**Funding acquisition:** Ku-Chung Chen, Chwen-Ming Shih, Chin-Cheng Lee.

**Investigation:** Chwen-Ming Shih, Chih-Ming Chou, Chia-Hsiung Cheng, Chin-Cheng Lee.

**Methodology:** Kuo-Hao Ho, Chih-Ming Chou.

**Supervision:** Chwen-Ming Shih, Chih-Ming Chou, Chia-Hsiung Cheng.

**Validation:** Peng-Hsu Chen, Kuo-Hao Ho, Chin-Cheng Lee.

**Writing – original draft:** Peng-Hsu Chen, Kuo-Hao Ho.

**Writing – review & editing:** Ku-Chung Chen, Chin-Cheng Lee.

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
