## [Decision Letter · Decision Letter 0]

22 Aug 2019

PONE-D-19-20240

IGF-1-enhanced miR-513a-5p signaling desensitizes glioma cells to temozolomide by targeting the NEDD4L-inhibited Wnt/β-catenin pathway

PLOS ONE

Dear Dr. Lee,

Thank you for submitting your manuscript to PLOS ONE.  I have received to date one completed review and could not get a second one. I have personally evaluated your article and I totally agree with the comments raised by the reviewer. Based on these recommendations, and after careful consideration, we feel that it has merit but does not fully meet PLOS ONE’s publication criteria as it currently stands. Therefore, we invite you to submit a revised version of the manuscript that addresses the points raised during the review process.

Specifically, the reviewer has found that your study contains interesting findings but also underlined significant weaknesses in the methodology. You should precisely answer to his 10 distinct comments. A particular attention will be paid to the concerns related to methodological and statistical issues as well as interpretation of the data. I also think that the use of a loss of-function approach with anti-miR would be instrumental to assess the function of the endogenous miRNA on TMZ sensitivity as well as on its target. Regarding the last point raised by the reviewer (glioma stem cells), while I think the use of patient-derived glioma stem cells would strengthen the scope of the study, it will not represent an essential criterion for the final decision.  Finally, I have also personally raised one additional point: while the authors state that all data are fully available, they do not provide a link for the microarray dataset. The whole miRNA microarray dataset should be deposited to a public database such as Gene Expression Omnibus (https://www.ncbi.nlm.nih.gov/geo/). 

We would appreciate receiving your revised manuscript by Oct 06 2019 11:59PM. To enhance the reproducibility of your results, we recommend that if applicable you deposit your laboratory protocols in protocols.io, where a protocol can be assigned its own identifier (DOI) such that it can be cited independently in the future. For instructions see: http://journals.plos.org/plosone/s/submission-guidelines#loc-laboratory-protocols

We look forward to receiving your revised manuscript.

Kind regards,

Bernard Mari, Ph.D

Academic Editor

PLOS ONE

**Journal Requirements:**

**Comments to the Author**

1. Is the manuscript technically sound, and do the data support the conclusions?

Reviewer #1: Partly

2. Has the statistical analysis been performed appropriately and rigorously? 

Reviewer #1: Yes

3. Have the authors made all data underlying the findings in their manuscript fully available?

Reviewer #1: Yes

4. Is the manuscript presented in an intelligible fashion and written in standard English?

Reviewer #1: Yes

5. Review Comments to the Author

Reviewer #1: In this article the authors have identified in GBM cell lines, a novel IGF1 dependent molecular mechanism in where miR-513a-5p represses NEDD4L leading to activation of the WNT/beta-Catenin pathway. This beta-Catenin activation would confer higher resistance to TMZ treatment. The study is interesting but needs more development before to jump too fast at the conclusion. See specific comments below.

Comments to the authors :

In the figure 1, the authors need to clearly demonstrate beta-catenin nuclear translocation upon IGF1 treatment. This could be done by immunofluorescence or by cell fractionation. Also showing beta-Catenin nuclear localisation is not enough to conclude to an increase of its activation upon IGF1. The authors need to assess its transcriptional activity upon IGF1 treatment by using a reporter gene containing TCF/LEF binding sites (top flash reporter gene).

The cell proliferation has been determine by MTT assay which reveals only 13% increase of cell proliferation upon IGF1. This is an indirect method to measure cell proliferation which need to be confirmed by counting viable cells following IGF1 treatment.

Figure 1 shows an increase of cyclin D1 expression upon IGF1 treatment. Although cyclin D1 is a well known target of beta-Catenin many other factors may stimulate this gene upon growth factor stimulation. The authors have to show that this increase is beta-Catenin dependent by performing its functional invalidation using specific siRNA.

Similar functional invalidation need to be conducted to assess whether or not the 13% increase of proliferation is due to WNT/beta-Catenin activation.

Cell sensitivity to TMZ mediated death has been carried out by using MTT assay. A direct method to measure cell death will be much more appreciated.

Beta-Catenin invalidation need to be performed to confirm its contribution in TMZ resistance. This will also confirm the results obtained with GSK3-beta inhibition which is not functionally obvious in the case of M059K cells. The quantification of DNA damages would be also informative in this context.

The authors show increased miR-513a-5p upon IGF1 stimulation. It would be interesting to assess the effect of miR-513a-5p invalidation (anti-miR) on GBM cell sensitivity to TMZ following IGF1 treatment.

The results clearly indicate that NEDD4L 3’UTR can be bound by miR-513a-5p. MiR-513a-5p overexpression induces the decrease of NEDD4L protein. These results are convincing but based on gene overexpression. In order to confirm the physiological relevance of these results, the authors need to perform miR-513a-5p invalidation (upon IGF1 treatment) and evaluate NEDD4L expression.

The consequences of miR-513a-5p invalidation on beta-Catenin nuclear localization need also to be assessed to confirm the relevance of this miR for activation of this pathway.

To validate their study, the authors have to reproduce at least part of their data in patient-derived glioma stem cells which are definitely much more relevant to study GBM physiology than GBM cell lines.

6. PLOS authors have the option to publish the peer review history of their article (what does this mean?). If published, this will include your full peer review and any attached files.

Reviewer #1: Yes: Thierry Virolle

---

## [Author Response · Author response to Decision Letter 0]

23 Oct 2019

Thank you very much for reviewing our article and providing instructive comments. We have tried our best to address your concerns/suggestions in the revised manuscript, and hopefully the revised manuscript can reach the publishable level. The point-by-point responses are shown in the attached file.

---

## [Editor Report · Decision Letter 1]

12 Nov 2019

PONE-D-19-20240R1

IGF-1-enhanced miR-513a-5p signaling desensitizes glioma cells to temozolomide by targeting the NEDD4L-inhibited Wnt/β-catenin pathway

PLOS ONE

Dear Dr. Lee,

Thank you for resubmitting your manuscript to PLOS ONE. While you have adequately addressed most of the queries in the review and that the revised manuscript is significantly improved from its original submission, one issue still needs to be addressed before full acceptance of the paper.

Specifically:

As mentioned during the first review, the authors should provide a link for the microarray dataset. The whole miRNA microarray dataset should be deposited to a public database such as Gene Expression Omnibus (https://www.ncbi.nlm.nih.gov/geo/) and the GSE reference should be included in the material and method section.

We would appreciate receiving your revised manuscript by Dec 27 2019 11:59PM. To enhance the reproducibility of your results, we recommend that if applicable you deposit your laboratory protocols in protocols.io, where a protocol can be assigned its own identifier (DOI) such that it can be cited independently in the future. For instructions see: http://journals.plos.org/plosone/s/submission-guidelines#loc-laboratory-protocols

We look forward to receiving your revised manuscript.

Kind regards,

Bernard Mari, Ph.D

Academic Editor

PLOS ONE

---

## [Author Response · Author response to Decision Letter 1]

13 Nov 2019

Thanks for the suggestion. We uploaded the array data to GEO database as GSE140297. All the information will be released on Nov 14, 2019. We add and highlight this information in the method section of revised manuscript.

---

## [Editor Report · Decision Letter 2]

15 Nov 2019

IGF-1-enhanced miR-513a-5p signaling desensitizes glioma cells to temozolomide by targeting the NEDD4L-inhibited Wnt/β-catenin pathway

PONE-D-19-20240R2

Dear Dr. Lee,

We are pleased to inform you that your manuscript has been judged scientifically suitable for publication and will be formally accepted for publication once it complies with all outstanding technical requirements.

With kind regards,

Bernard Mari, Ph.D

Academic Editor

PLOS ONE
---

## [Editor Report · Acceptance letter]

21 Nov 2019

PONE-D-19-20240R2 

IGF-1-enhanced miR-513a-5p signaling desensitizes glioma cells to temozolomide by targeting the NEDD4L-inhibited Wnt/β-catenin pathway 

Dear Dr. Lee:

I am pleased to inform you that your manuscript has been deemed suitable for publication in PLOS ONE. Congratulations! Your manuscript is now with our production department. 

With kind regards,

on behalf of

Dr. Bernard Mari 

Academic Editor

PLOS ONE